# Biophysical Gradients and Performance of Whitebark Pine Plantings in the Greater Yellowstone Ecosystem

**David Laufenberg [1],\*, David Thoma [2], Andrew Hansen [1] and Jia Hu [3]**

1   Department of Ecology, Montana State University, 310 Lewis Hall, P.O. Box 173460,
    Bozeman, MT 59717, USA; hansen@montana.edu
2   National Park Service Inventory and Monitoring Program, 2327 University Way, Bozeman, MT 59717, USA;
    dave_thoma@nps.gov
3   School of Natural Resources and the Environment, University of Arizona, 1064 E. Lowell Street,
    Tucson, AZ 85721, USA; jiahu@email.arizona.edu
*   Correspondence: david.laufenberg@gmail.com; Tel.: +1-608-335-3099

**Abstract:** *Research Highlights:* The efficacy of planting for restoration is important for ecosystem managers. Planting efforts represent an opportunity for conserving and managing species during a population crisis. *Background and Objectives:* Federal agencies have been planting whitebark pine (WBP), an important subalpine species that is late to mature and long-lived, for three decades in the Greater Yellowstone Ecosystem (GYE). These efforts have been met with varying success, and they have not been evaluated beyond the first five years post-planting. Ecosystem managers will continue to plant WBP in the GYE for years to come, and this research helps to inform and identify higher quality habitat during a period of changing climate and high GYE WBP mortality rates. *Materials and Methods:* We use a combination of field sampling and a water balance model to investigate local biophysical gradients as explanatory variables for WBP performance at twenty-nine GYE planting sites. *Results:* We found that the WBP growth rate was positively correlated with actual evapotranspiration (AET) and was greatest when cumulative growing season AET was above 350 mm. Growth rate was not strongly affected by competition at the levels found in this study. However, site density change over time was negatively affected by mean growing season temperature and when more than five competitors were present within 3.59 m radius. *Conclusions:* If they make it to maturity, trees that are planted this season will not begin to produce cones until the latter half of this century. We recommend planting efforts that optimize AET for growth rate objectives, minimize water deficit (WD) that cause stress and mortality, and removing competitors if they exceed five within a short distance of seedlings.

**Keywords:** forest management; wildland health; whitebark pine; water balance

---

## 1. Introduction

The Greater Yellowstone Ecosystem (GYE) is often considered to be the largest remaining, nearly-intact ecosystem in the contiguous United States. Whitebark pine (*Pinus albicaulis*; WBP) is widely recognized as a keystone and foundation species in the high country of the GYE, and over half of the aerial extent of WBP in the United States is found there [1]. Notably, nearly half of the GYE WBP distribution has exhibited severe mortality in recent years [2], due to higher temperatures and drier conditions and the associated insect outbreaks and an introduced pathogen [3–6]. A recent analysis indicates that 51% of all standing WBP (>22.8 cm diameter breast height) in the United States (US) are dead [7]. Grizzly bears (*Ursus arctos*, Rausch) are federally listed as threatened under the Endangered Species Act (ESA), in part due to impact from the loss of WBP seeds, which represent an important

food source [8]. Threats to WBP are imminent, of high magnitude, and, thus, the sp'ecies warrants protection under the ESA [9].

As part of a strategy to conserve the species, the United States Department of Agriculture's Forest Service - Northern Region incorporates climate change adaptation strategies into management actions based on the vulnerability of resources to climate change [10]. WBP has been identified as the most vulnerable tree species in its range to climate change [11], and future climate change is projected to dramatically reduce habitat suitability [12]. Additionally, WBP is slow to reach maturity and can live for more than 1000 years [13]. The effects of climate influence life-history processes such as seedling establishment and growth well before noticeable biogeographic range shifts occur [14] and in this long-lived species, the seedling life stage is likely the most sensitive to environmental factors [15]. Therefore, it is important to consider both current and future climate when evaluating the management treatments, such as where to plant seedlings [1].

It is possible that recruitment in historically favorable sites will no longer facilitate passage from seedling to adult, given climate change and forecasts for continued warming and drought events [15,16]. Climate niche and species distribution models often assume that young individuals have similar tolerances as adult trees [12], but this is unlikely because seedlings with limited root systems, low carbon reserves, and reduced photosynthetic capacity may perish under conditions that present no difficulty for conspecific adults [15]. Although some GYE WBP are likely water stressed by late summer and fall similar to WBP in other areas of the West [17], this has not been documented in the GYE [18]. At some point in the past, adult trees were young at the same site, so the conditions were amenable to their development. Periods of actual evapotranspiration when solar energy and soil moisture are simultaneously available represent the timing, duration, and magnitude of opportunities for growth are represented, which allows for transpiration and growth. Conversely, deleterious effects of water stress occur when temperatures are warm enough for growth, but water is insufficiently supplied by the soil [19]. Climate conditions that are suited to growth can be estimated by tracking actual evapotranspiration and conversely water stress can be estimated by periods of water deficit, which is water need that is not met by water available in the soil [19]. Actual evapotranspiration and water deficit can both be estimated while using water balance models that account for input and output of water from soil.

The locations of GYE restoration planting sites are generally consistent with the WBP planting guidelines [10] where regeneration efforts take advantage of post-burn habitat by planting WBP seedlings in burned WBP stands prior to site occupation by potential competitors [20]. The guidelines presume that, if WBP previously existed in these locations, they may support WBP in the future [21]. However, climate in the future might move outside historical ranges and might already present challenges to seedling establishment [15,16]. It is not currently known how the range of climate and soil conditions in these planting sites relates to water availability and seedling growth and survival.

While the loss of some seedlings may be due to disease, physical damage from wildlife trampling or snow, competition might also play an important role in establishment [21,22]. Below treeline in the GYE, which is near the species' southern range of its distribution [23], the effect of competitors on WBP growth is negative [22], rather than facilitative, as has been described near treeline where other tree species provide protective cover [21]. Below treeline competition generally has an inverse relationship with WBP performance [10]. The negative effects of competition could result from shading, reduced access to nutrients, or soil moisture [21,24]. Regardless of the mechanism, competition complicates a purely climate-based understanding of establishment and growth. The restoration efforts underway provide an opportunity to study survival and growth rates in wildland settings that include gradients of competition and climate.

This study investigates the biophysical conditions that affect seedling survival and growth that, in part, determine establishment, which is a necessary precursor to reproduction. We were interested to learn if climate or competition were correlated with variation in survival rates that could be used to inform planting guidelines in the future. Thus, local analyses of the correlates with WBP seedling

survival is an opportunity to leverage spatial variability within and between existing restoration planting sites to improve the understanding of competition and water relationships conducive to establishment and survival. Specifically, we evaluated survival and growth rate, as related to water balance and competition at the tree level across a large environmental gradient where seedlings have been planted since 1990.

## 2. Materials and Methods

### 2.1. Study Site

The GYE is approximately 10.1 million ha (25 million acres), which includes two national parks, five national forests, two national wildlife refuges, two tribal reservations, and a number of state and privately managed lands. Over the past four decades, the United States Forest Service and the National Park Service have established approximately 1500 acres of planted WBP in the GYE. There are five planting units in this study (Figure 1), with each containing between two and eight planting sites.

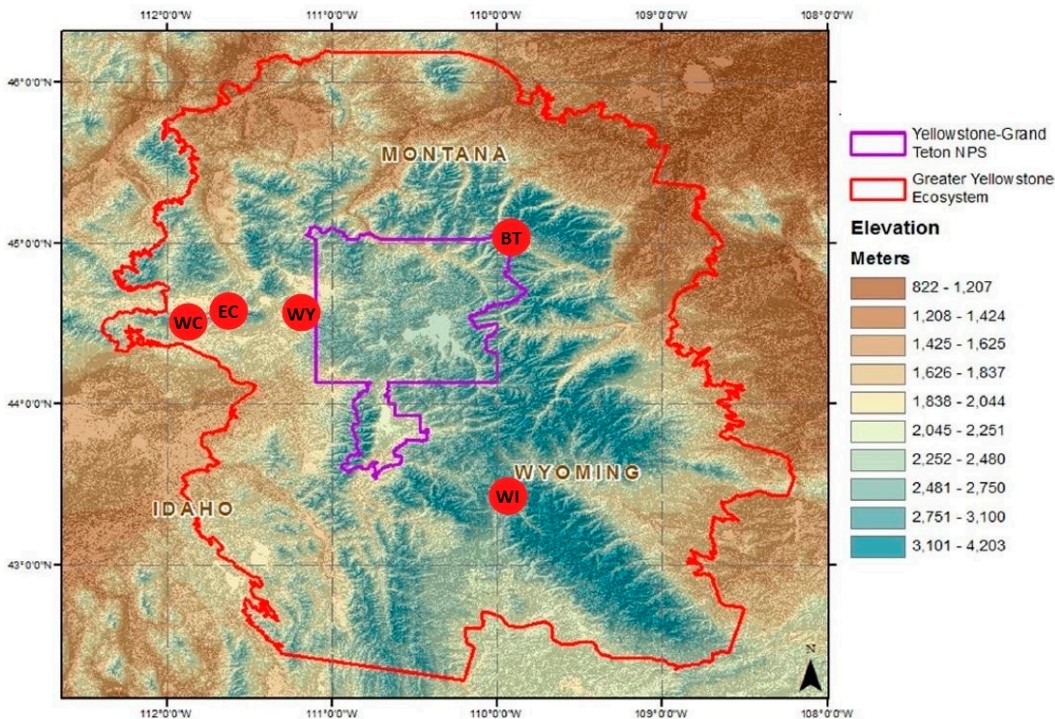

**Figure 1.** Greater Yellowstone Ecosystem with whitebark pine planting units (red dots with black two-letter planting unit abbreviations; BT = Beartooth, EC = East Centennial, WC = West Centennial, WI = Wind River, WY = West Yellowstone) and Yellowstone and Grand Teton national park boundaries (modified from Chang et al. 2014).

### 2.2. Sampling Design

Our hierarchical sampling design included five planting units, 29 planting sites in total within those units, and hundreds or thousands of WBP seedlings planted at each site (Figure 2). The sites were sampled from May 2018 to October 2018 while using a random starting point, then systematic sampling of every 20th grid cell from a matrix of 10 m × 10 m grids overlaid on the planting site. The planting sites ranged from 0.41 to 23.2 ha 2% to 15% of each site was sampled, depending upon size and biophyisical heterogeneity. In general, larger sites had proportionately fewer trees sampled than smaller sites. Each WBP within every grid was digitally tagged (±3.5 m resolution). We used Survey123 software to collect field data at each WBP.

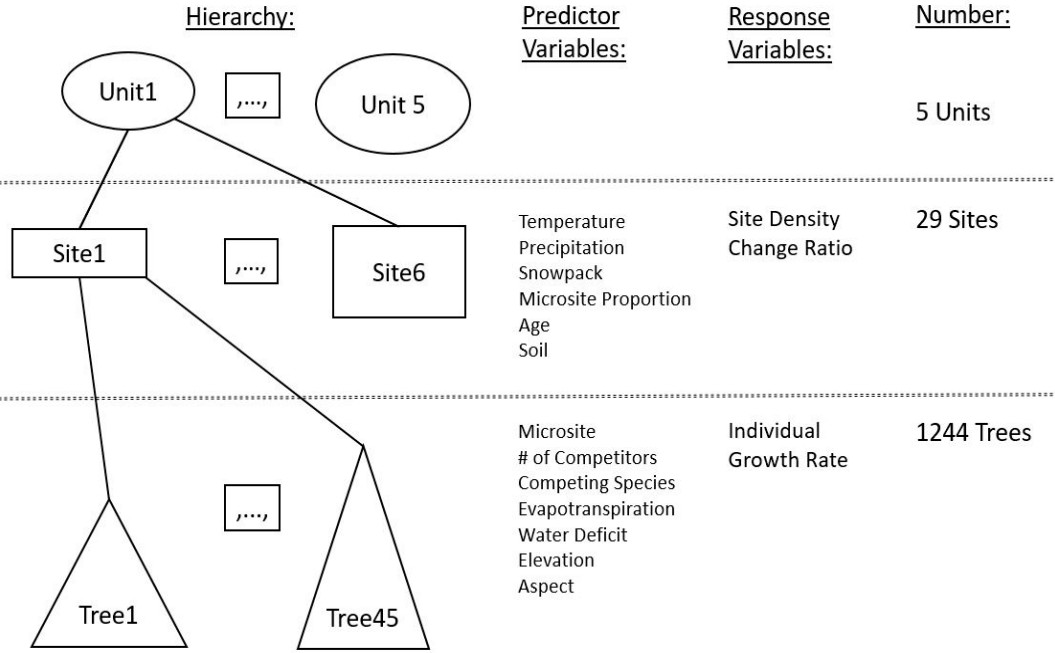

**Figure 2.** Hierarchical study design, predictor variables, and sample sizes.

### 2.3. Data

#### 2.3.1. Historical Data

We used planting records from the forest districts with WBP planting prior to 2013 (Table 1). The number of seedlings initially planted (as recorded in USDA Forest Service records) was divided by the planting site area to calculate the initial WBP density at each site (Table 2). At the time of planting, the height of the seedlings was 15.2 cm (6 inches) on average and the root wads were 20.3 cm (8 inches) long on average (personal communication with USDA Forest Service managers).

**Table 1.** Descriptions of planting units. Elevation, latitude, and longitude (NAD83 projection) is an approximate center representing the planting unit centroid.

| Planting Unit | Number of Planting Sites | Years Planted | Elevation (m) | Latitude (°) | Longitude (°) |
|---|---|---|---|---|---|
| Beartooth | 7 | 1991, 1993 | 2611 | 45.03 | −109.90 |
| East Centennial | 8 | 2010, 2012 | 2352 | 44.53 | −111.61 |
| West Centennial | 6 | 2011 | 2650 | 44.51 | −112.01 |
| West Yellowstone | 4 | 1998–2002 | 2408 | 44.47 | −111.13 |
| Wind River | 4 | 2002, 2013 | 2871 | 43.53 | −109.84 |

**Table 2.** Descriptions of planting sites (BT = Beartooth, EC = East Centennial, WC = West Centennial, WI = Wind River, WY = West Yellowstone). Initial planting density trees per hectare (TPH).

| Planting Site | Trees Planted | Hectares Planted | Density (trees/ha) |
|---|---|---|---|
| BT1 | 1700 | 1.59 | 1069 |
| BT2 | 1000 | 0.41 | 2447 |
| BT3 | 980 | 0.88 | 1116 |
| BT4 | 1000 | 1.06 | 943 |
| BT5 | 8335 | 7.86 | 1060 |
| BT6 | 3110 | 2.39 | 1300 |
| BT7 | 1000 | 1.19 | 838 |
| EC1 | 4835 | 11.34 | 426 |
| EC2 | 999 | 2.34 | 426 |

**Table 2.** *Cont.*

| Planting Site | Trees Planted | Hectares Planted | Density (trees/ha) |
|---|---|---|---|
| EC3 | 2896 | 6.79 | 426 |
| EC4 | 3603 | 4.38 | 823 |
| EC5 | 1078 | 1.28 | 840 |
| EC6 | 440 | 0.65 | 680 |
| EC7 | 1839 | 4.31 | 426 |
| EC8 | 3930 | 9.22 | 426 |
| WC1 | 3610 | 7.02 | 514 |
| WC2 | 1390 | 2.70 | 514 |
| WC3 | 2182 | 4.25 | 514 |
| WC4 | 2802 | 5.45 | 514 |
| WC5 | 4833 | 9.40 | 514 |
| WC6 | 6381 | 12.41 | 514 |
| WY1 | 5000 | 4.62 | 1083 |
| WY2 | 1100 | 1.80 | 611 |
| WY3 | 10,000 | 11.18 | 894 |
| WY4 | 2800 | 3.29 | 852 |
| WI1 | 7434 | 12.42 | 598 |
| WI2 | 5101 | 23.18 | 220 |
| WI3 | 3728 | 16.38 | 228 |
| WI4 | 3924 | 16.38 | 2447 |

2.3.2. Field Data

We documented the location and number of each living WBP, and its height (cm) within all of the sampled grids. Height was measured from the base of the tree to its highest point to the nearest centimeter. Subsequently, to characterize the potential competition within and among species, for each WBP sapling ($n = 1244$), we counted, identified, and measured the height of each conifer greater than 50 cm height within a circular radius of 3.59 m (1/100th acre).

Individual annual growth rates were determined by the change in present-day height relative to planted height divided by the years since planting. Additionally, 2.5 years were subtracted from the years since planting to account for the establishment period, whereby young seedlings invest sequestered carbon and reserves into root instead of shoot growth (discussions with USDA Forest Service managers). We use height as a proxy for growth, because trees were too small to core and destructive sampling for tree ring widths in the restoration planting sites would be counterproductive.

The WBP density change ratio at each site was determined by dividing the present-day WBP density by the initial WBP planting density. We estimated current WBP site density as the mean density of our sampled grids at each site. Per conversations with USDA forest service managers, planting sites were void of trees at the time of planting, but naturally germinating trees may have been present if the fire was low intensity, and post planting natural regeneration was also possible.

The presence or absence of a microsite was documented for each individual. A microsite was considered to be present when there was an object (rock or stump) of at least 5 cm width and 10 cm height within 20cm of the WBP that could modify the local environment by providing shade, shelter or increased water supply due to the capture of fine sediments or litter that help to retain moisture. We also measured aspect (0°, 45°, 90°, 135°, 180°, 225°, 270°, 315°, or flat) and slope (0°, 5°, 10°, 15°, 20°, or 25°) at each WBP while using a compass (11.5° E declination) and an inclinometer. Additionally, each WBP seedling was visually inspected for the presence of white pine blister rust.

Soil field capacity, or water holding capacity, is the amount of water soil can hold after it freely drains from a saturated state by gravity. Estimates of soil texture and volumetric rock content, including boulders, cobbles, stones >2 mm diameter that reduce water holding capacity of soil in the rooting depth, informed an estimate of soil field capacity for each tree. The soil texture was estimated using a texture-by-feel method that was based on moist soil ribbon length and grittiness. This is a repeatable

field-method for determining the textural class from a textural triangle of sand, silt, and clay fractions (Figures A1 and A2). We used a mean rooting depth of 203.2 mm (8 inches) per discussions with US Forest Service managers, in our estimations of soil field capacity (Table A1).

Shading at the individual level was estimated while using photography software (CanopyApp) on a smartphone (Apple iPhone 5s) with a wide-angle lens attachment (AMIR for iPhone 0.4X Wide Angle Lens). Canopy cover that is estimated by this method is repeatable, easy, and more accurate than traditional methods, such as a spherical densiometer [25]. A photo was taken from a position 1.37 m south and 1.37 m (same as Diameter Breast Height forestry standard) above ground height of each WBP to quantify the percent of shading and, therefore, a reduction in energy received in the water balance model (Figure 3).

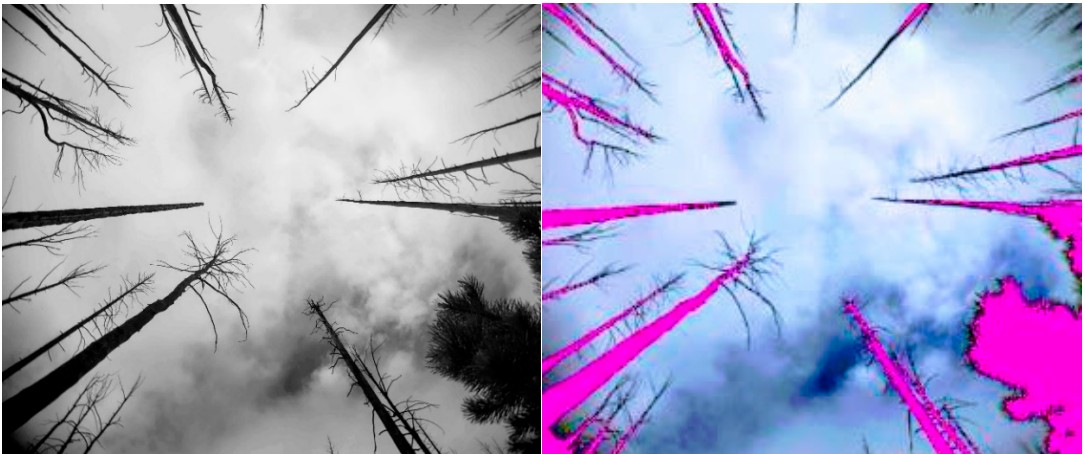

**Figure 3.** Example of CanopyApp software used for shading estimates at the individual tree level. Photo on the right includes the software's generated mask (pink). For example, this particular tree experienced 14.2% shading.

## 2.4. Climate and Water Balance

We used 1km gridded, daily temperature, and precipitation data from the 1980–2017 Daymet dataset [26] as input to a water balance model. We used a Thornthwaite-type water balance model [27–30] to estimate water balance at each tree. The model estimates monthly and annual cumulative growing degree days, potential and actual evapotranspiration (PET and AET), soil moisture, and water deficit (WD), as described by Stephenson [31]. These variables were used to analyze the relationships between water availability, use, and need with WBP performance metrics of growth rate and density change ratio. The base model was modified to account for heat load, which is the greatest on southwest facing slopes [32]. It uses the Penman–Monteith equations to estimate PET, which is largely influenced by temperature, solar radiation, humidity, and wind (Table 3). Heat load and shade modifiers (values between zero and one) were used to adjust PET down or up, depending on slope, aspect, and shade measured at each tree. The model was then further modified by varying water holding capacity due to soil texture and coarse fragment content to calculate and distinguish between periods of accumulating WD and AET (Table 4). Relationships that focused on growing season (April–October) were explored, given that saplings are completely covered by snowpack during the winter months. The growing degree days were modeled per McMaster [33] with a temperature base of 5 °C.

**Table 3.** Potential evapotranspiration water balance model modifiers.

| Water Balance Variable | Relationship (+/−) with PET |
|---|---|
| Temperature (°C) | + |
| Elevation (m) | − |
| Aspect (°) | +/− (45° is the lowest, 225° is the highest) |
| Shading (%) | − |
| Slope (°) | +/− (minor impact, and dependent upon aspect) |

**Table 4.** Actual evapotranspiration water balance model modifiers.

| Water Balance Variable | Relationship (+/−) with AET |
|---|---|
| Soil Texture (% of sand, silt, clay) | +/− (dependent upon textural triangle) |
| Coarse Fragment (%) | − |
| Temperature (°C) | +/− (dependent upon elevation) |
| Precipitation (mm) | +/− (dependent upon elevation) |

*2.5. Analysis*

We identified the correlates with WBP performance (Table 5) by comparing multiple mixed effects models while using fixed effects, interactions, linear and polynomial functions, and different variance structures [34]. Models were assessed through the use of the corrected Akaike Information Criterion for finite sample sizes (AICc; [34]). We tested for collinearity between variables with a cutoff of ±0.6. We chose a parsimonious set of water balance variables by selecting the more biologically relevant variable from variable pairs that were correlated beyond our cutoff. In addition to the fixed effects of PET, AET, and WD, we also investigated the models that included the number of competitors, age of the planting site, and microsite presence/absence. Statistical software "R" was used for analyses, and mixed-effects modelling used the "lme4" package [35].

**Table 5.** Predictor variables and their definitions for use in mixed effects models. All of the climate variables were calculated based on the years since planting at that particular site.

| Predictor Variable | Definition |
|---|---|
| Age | Years since planting |
| $T_{mean}$ (°C) | Mean annual temperature |
| $T_{max}$ (°C) | Maximum monthly temperature |
| PPT (mm) | Mean annual precipitation |
| Snowpack (mm) | Mean spring (March–May) snowpack |
| Rain (mm) | Mean spring (March–May) rain |
| $WD_{annual\_mean}$ (mm) | Mean annual water deficit (April–October) |
| $WD_{annual\_max}$ (mm) | Maximum annual water deficit (April–October) |
| $WD_{month\_max}$ (mm) | Maximum monthly water deficit (April–October) |
| PET (mm) | Mean potential evapotranspiration (April–October) |
| AET (mm) | Mean actual evapotranspiration (April–October) |
| GDD | Mean annual growing degree days (April–October) |
| Comp_number | Number of conifers within 3.59 m radius of WBP |
| PIEN | Presence of *Pinus engalmanii* within 3.59 m radius of WBP |
| ABLA | Presence of *Abies lasiocarpa* within 3.59 m radius of WBP |
| PICO | Presence of *Pinus contorta* within 3.59 m radius of WBP |
| Micro | Microsite presence or absence at the individual-level |
| $Micro_{prop}$ | Proportion of WBP with a microsite at the site-level |

We related growth rate to our predictor variables after normalization via natural log transformation. Quantile–quantile (QQ) plots indicated a slight-left skew of our data. By including a random effect for unit, we also accounted for the spatial autocorrelation of our data. Biological relevance was unclear when deciding to include either the number of competitors or the age of the planting site, which were

correlated (0.64). We chose to include the number of competitors, because one of our objectives was to investigate the effect of competition on WBP performance.

Similar to growth rate, we normalized the density change ratio at each site ($n = 29$) via natural log transformation. We characterized our sites by averaging the water balance output across trees in each site. The site-level proportion of competitors, number of competitors, and proportion of individual WBP with microsites were used as continuous variables for analysis.

## 3. Results

### 3.1. Individual Growth Rate

The individual WBP growth rates ranged from near zero cm year$^{-1}$ to 25 cm year$^{-1}$ across all units. The West Yellowstone, Wind River, and Beartooth Planting Units had the highest mean and maximum growth rates, but there was a large amount of variation within all units (Figure 4). The East Centennial and West Centennial Planting Units had lower growth rates, but they were also younger units (six years post-planting relative to the mean of 15 years across all sites) and, therefore, did not have as many seasons to grow and exhibit variation in growth rate.

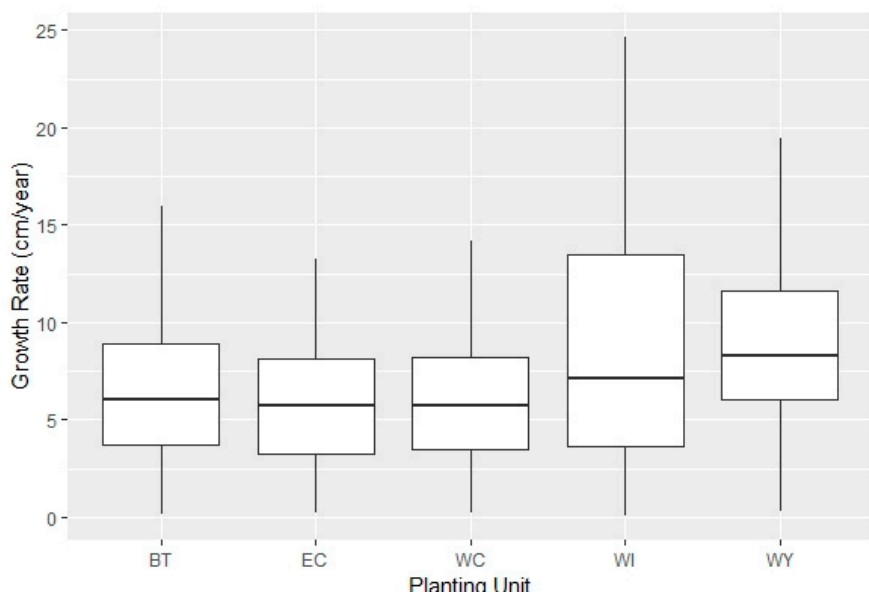

**Figure 4.** Adjusted growth rates (cm year$^{-1}$) across units (BT = Beartooth, EC = East Centennial, WC = West Centennial, WI = Wind River, WY = West Yellowstone).

The leading model for growth rate included AET and the number of local competitors. Our results indicated generally increasing growth rates as AET increased and weak evidence of decline in growth rate when there were more than 20 competitors within a circular radius of 3.59 m. At high levels of AET (>300 mm), the growth rate dramatically increased. The best supported model for individual growth rate (AICc = 3186.57, K = 9) included a cubic functional form for AET (*p*-value < 0.001) and number of competitors (*p*-value = 0.031; Table 6). Growth rate was positive at low levels of AET (<225 mm), but the relationship flattened from approximately 225 mm–350 mm of AET, and then was once again positive when the AET was greater than 350 mm ($\hat{\beta}_{AET} = 6.04$, SE = 1.47; $\hat{\beta}_{AET2} = 3.09$, SE = 0.98; $\hat{\beta}_{AET3} = 2.47$, SE = 0.94; Figure 5). The WBP growth rate was slightly positive at lower numbers of competitors, but at high numbers of competitors (>15 within 3.59 m radius), there was a negative relationship with growth rate of WBP ($\hat{\beta}_{Comp} = 2.38$, SE = 1.23; $\hat{\beta}_{Comp2} = -0.36$, SE = 0.96; $\hat{\beta}_{Comp3} = -1.82$, SE = 0.91; Figure 5).

**Table 6.** Mixed effects models of individual growth rate (AET = actual evapotranspiration during the growing season, PET = potential evapotranspiration during the growing season, PPT = mean annual precipitation, T = mean annual temperature, Micro = microsite presence/absence, Comp = number of competitors within 3.59m radius, PICO = *Pinus contorta* presence or absence within plot, PIEN = *Picea Engalmanii* presence or absence within plot, ABLA = *Abies lasiocarpa* presence or absence within plot).

| Individual Growth Rate Models | AICc | K |
|---|---|---|
| Null Model | | |
| Log(growth_rate) ~ 1 + random (Unit) | 3225.31 | 3 |
| Full Model | | |
| Log(growth_rate) ~ AET + PET + PPT + T + Micro + Comp_number + PICO + PIEN + ABLA + random (Unit) | 3262.92 | 12 |
| Best Model | | |
| Log(growth_rate) ~ $AET^3$ + Comp_number$^3$ + random (Unit) | 3186.57 | 9 |

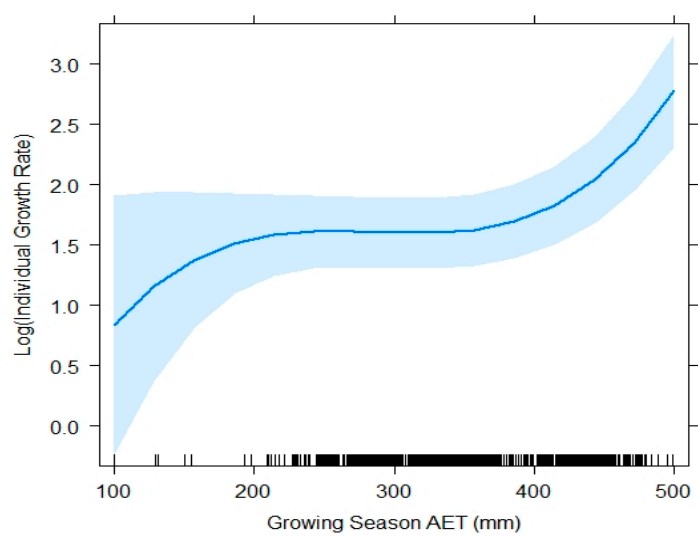

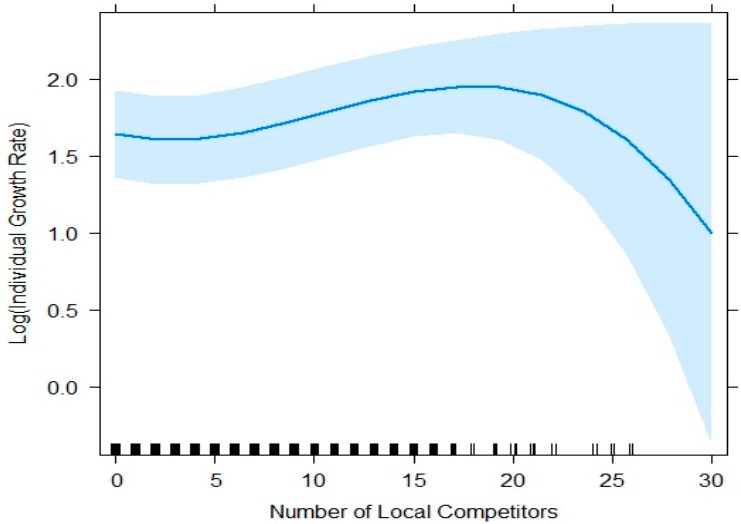

**Figure 5.** Plotted fixed effects of the best model for log (Individual Growth Rate) ~ $AET^3$ + Number of Local Competitors$^3$ + random (Unit) with confidence interval = 0.95 in light blue.

### 3.2. Site Density Change Ratio

The site density change ratios ranged from 0.1 to 4.2 across all units (Figure 6). The Wind River Planting Unit had the highest mean and maximum density change, and it was the only unit with higher present-day density of WBP relative to the initial planting density (ratio > 1 averaged across sites in each unit). Within and across units there was a large amount of variation between sites, but five of the 29 sites had density change ratio > 1, which indicated that natural regeneration outpaced the death of planted trees in those places (Wind River = 3, Beartooth = 1, and West Yellowstone = 1). Seven of the top eight performing sites were from the Beartooth or Wind River Planting Units.

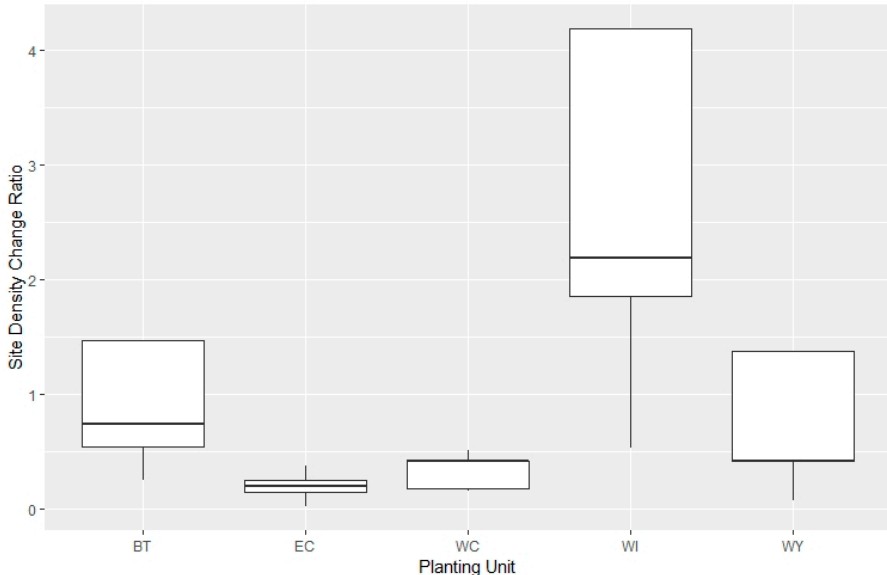

**Figure 6.** Density change ratio among sites and across units (BT = Beartooth, EC = East Centennial, WC = West Centennial, WI = Wind River, WY = West Yellowstone).

The leading model for site density change ratio included the monthly maximum temperature and mean number of local competitors. A greater number of seedlings survived between mean growing season maximum temperature of 14 and 15 °C when mean number of local competitors were less than seven. The best supported model for site density change ratio (AICc = 72.42, K = 9) included a cubic functional form for $T_{max}^3$ and number of competitors (Table 7). At lower temperature maximums, the relationship with site density change ratio was positive, but, as maximums increased, a negative relationship developed ($\hat{\beta}_{Tmax}$ = −2.49, SE = 1.19; $\hat{\beta}_{Tmax^2}$ = −0.71, SE = 1.00; $\hat{\beta}_{Tmax^3}$ = 0.62, SE = 0.93; Figure 7). At low numbers of competitors, there was a slightly positive relationship, but there was a generally negative relationship with site density change ratio of WBP ($\hat{\beta}_{Comp}$ = 2.34, SE = 1.28; $\hat{\beta}_{Comp^2}$ = −2.16, SE = 1.00; $\hat{\beta}_{Comp^3}$ = 1.35, SE = 1.11; Figure 7) at higher numbers of competitors (>5 within 3.59 m radius). During surveys, we found signs of white pine blister rust in only three of 1244 WBP trees sampled, including two trees in the Beartooth Planting Unit and one tree in the West Centennial Planting Unit.

**Table 7.** Mixed effects models of site density change ratio (AET = mean annual cumulative actual evapotranspiration, $WD_{max\_month}$ = maximum monthly water deficit, $T_{max}$ = maximum monthly site temperature, $Micro_{ratio}$ = ratio of microsite presence/absence, Comp = mean number of competitors within 3.59m radius, PICO = ratio of *Pinus contorta* presence or absence within plot, PIEN = ratio of *Picea Engalmanii* presence or absence within plot, ABLA = ratio of *Abies lasiocarpa* presence or absence within plot).

| Site Density Change Ratio Models | AICc | K |
|---|---|---|
| Null Model | | |
| Log(density_change_ratio) ~ 1 + random (Unit) | 83.72 | 3 |
| Full Model | | |
| Log(density_change_ratio) ~ AET + $WD_{max\_month}$ + $T_{max}$ + Microratio + Comp_number + PICO + PIEN + ABLA + random (Unit) | 97.80 | 11 |
| Best Model | | |
| Log(density_change_ratio) ~ $T_{max}^3$ + Comp_number$^3$ + random (Unit) | 72.42 | 9 |

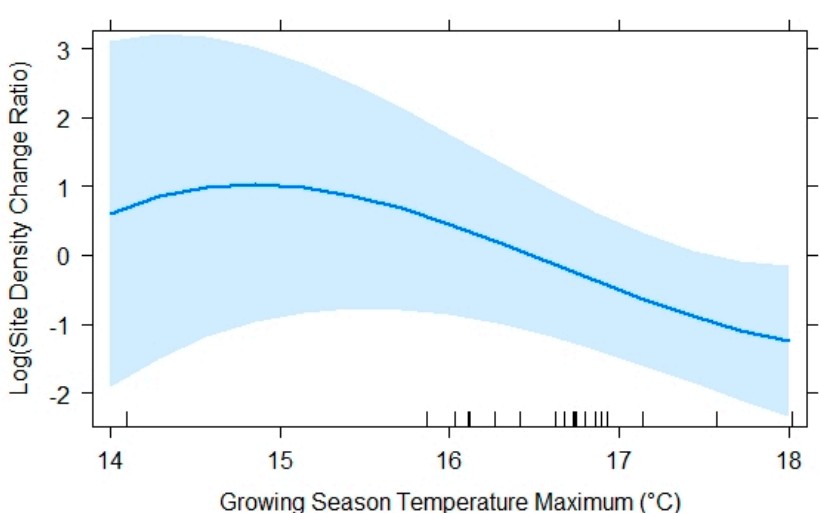

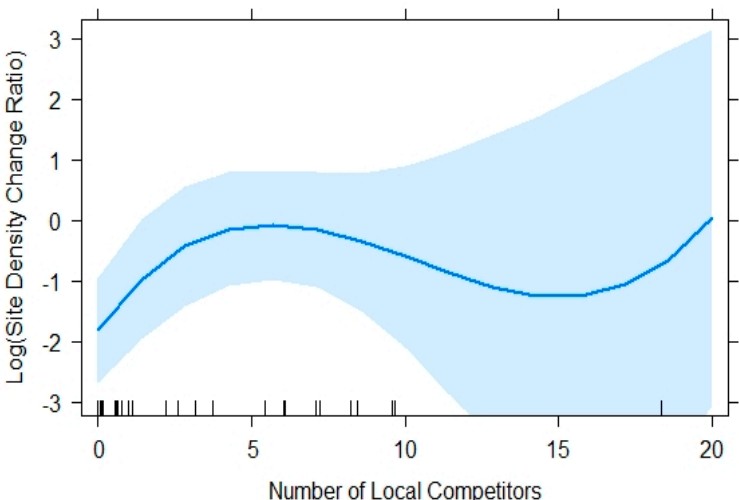

**Figure 7.** Plotted fixed effects of the best model for log(Site Density Change Ratio) ~ Growing Season Temperature Maximum$^3$ + Number of Local Competitors$^3$ + random(Unit) with confidence interval = 0.95 in light blue.

## 4. Discussion

### 4.1. Individual Growth Rate Models

The physiological basis for the growth response to AET is that it estimates the simultaneous length of time and magnitude of water and energy that are available for plant growth [19]. Specifically, AET is dependent on soil moisture and exponentially decreases as soil dries. The rate of soil moisture depletion is determined by the energy that is available to drive transpiration that varies with solar radiation interacting with slope, aspect, elevation, temperature, growing season length determined by snow cover, and day length. Therefore, it alleviates the need to speculate on complex combinations of precipitation and site characteristics as the determinants of growth rate, because these are quantitatively factored into the calculation of AET by the model.

This leads to an important finding that not all sites conducive to establishment and growth are equal even if they have similar temperature and precipitation regimes. Where, managers intentionally avoid planting in dry sites to avoid drought stress it is less commonly recognized that there is a wide range of environmental suitability for planting sites in the GYE that range between 200 and 500 mm AET during the growing season. Importantly, growth rate is associated with water use that is estimated by AET, which integrates climate and site characteristics quantitatively to estimate a fraction of total annual precipitation that is used by plants. If the goal is to plant trees in sites where growth rate is fastest, managers could target sites where the annual AET rates exceed 350mm in the GYE. Alternatively, if the goal is to plant trees that reach maturity at different times in the future to mitigate against insect epidemics, they could use the results from this study to plant in a range of locations where growth rates are suitable but would diversify timing of reproductive potential and susceptibility to insect attack [18]. However, it should be noted that the effects of climate differ with life stage, so the effects on seedlings may be different in older life stages [36]. These findings suggest focus on AET using a water balance approach can simplify identification of suitable planting sites if the focus is on management for growth rate.

We did not find strong evidence that competition at the most common levels we measured caused a reduction in growth rate over time since planting, consistent with findings from a study of growth release from thinning in whitebark pine [37]. These are still relatively open sites with little shading due to their early successional stage after fires and short stature of planted trees and most competitors. However, we found weak support (confidence interval includes zero) for declines in growth rate at the highest levels of competitor density in our study. Notably, these were also older sites (>15 years since planting). A decline in growth rate due to conifer competition was consistent with findings below treeline from a study on Mt. Rainier, WA [21]. Our findings provide some support for thinning competitors at older sites with higher numbers of competitors (>15 competitors within 3.59m radius) to release WBP growth, but the benefits may not be realized until the WBP trees are older and more of the competitor canopy closes and shades WBP [37].

### 4.2. Site Density Change Ratio Models

Our finding that stand density decreased with increasing maximum growing season temperature is consistent with the findings in energy limited environments, where temperature was as good a predictor of mature tree mortality as water deficit. In such environments, water stress is less common than in water limited environments and, when water stress does occur, it might be small in magnitude [38]. Thus, the basis for reduction in WBP density related to temperature might be due to its direct effects on growing season water availability and physiology or indirect relationships between insects and pathogens that can be more destructive with warmer temperatures, all of which can contribute to mortality [39]. This finding is also consistent with a comprehensive review that found increasing temperatures and lower moisture availability in post-fire sites after 2000 reduced tree regeneration in the subalpine forests of the U.S. Rocky Mountains [40]. Although water deficit did not appear in our top model, it is positively related with temperature and might not have been

an important correlate in our study if the water deficit levels since planting were not high relative to seedling tolerance.

The inflection of change density ratio at five competitors suggests a threshold, where less than five competitors within 3.59 m radius may actually be beneficial for young WBP trees, because they may provide microsite advantages that could modify the local environment by providing moderate shade reducing evaporation from soil, shelter, or increased water supply due to the capture of fine sediments or litter that help to retain moisture. However, when more than five competitors were present, they likely outcompete WBP for water or nutrients in the seedling stage of development, resulting in the loss of WBP [10,21]. It is less likely at this early successional stage that competition for light was the explanation for the reduction in change density ratio, because 86 percent of sampled WBP were less than 20 percent shaded, which confirmed that seedlings were not heavily shaded. For this reason, we conclude that water or nutrient limitations more likely explain the competitor effect. However, revisits in the future could clarify the effects of increasing shade if competitor growth outpaces WBP growth and the effect becomes more apparent.

### 4.3. Water Deficit

The West Yellowstone Planting Unit had the highest growth rates, but in three of four sites a change ratio less than one meaning mortality outpaced natural regeneration. This seemingly contradictory finding is consistent with a physiological explanation of water balance influence on vegetation condition, where multiple aspects of water balance influence vegetation condition. Although the West Yellowstone Planting unit receives the highest annual precipitation, the high water deficit was driven by low water holding capacity of rhyolitic soil in the shallow rooting zone and its southerly aspect, which promotes more rapid evapotranspiration and late season water deficit in the absence of rain. In other words, rapid growth can be maintained when precipitation input sustains soil moisture for evapotranspiration. However, if summer precipitation is not sustained, deficit can accumulate rapidly, which might explain the mortality and change ratio less than one due to the low water holding capacity of rhyolitic soil. This points out the value of a water balance approach that illuminates the complex interactions of climate and site conditions.

Another study found that annual moisture deficits were significantly greater in 2000–2015 relative to the 1985–1999 period in the U.S. Rocky Mountains [40]. This is believed to contribute to failure of regeneration post-fire, which has been noticed at higher levels in recent decades in the U.S. Rocky Mountains [40]. This might also explain why north-facing aspects have been shown to exhibit greater regeneration than south-facing aspects within burned landscapes [41].

A consistent interpretation of dryness is that it makes trees more susceptible to fire [22]. At the same time, mature WBP can survive dry sites, where other trees cannot, and in these places fire keeps other competitors at low levels once WBP are mature [24]. For these reasons, the concept of dryness as a driver of WBP life history and future distribution is critically important, but should not be the only climate factor that is considered in WBP management. As shown here, actual evapotranspiration is also important to establishment and growth.

Forests that occur along the climatically dry periphery of their range are more susceptible to conversion to non-forest due to increasing trends in annual moisture deficits lending to decreases in tree regeneration post-wildfire [40]. This is a major consideration for the restoration of a species that requires at least half a century of growth prior to production of seeds. Higher WD exhibited at the West Yellowstone Planting Unit is the likely "dry periphery" of WBP range in the GYE. Indeed, this area is part of the species' range edge [42]. Notably, convergent hillslope positions can maintain shallow, subsurface water flows that augment water supply in otherwise dry climates, even in areas of high WD [43]. That is, a potential planting site is not simply good or bad, but among the good sites, some are clearly better. Those that are not optimum could still be considered if planting is focused on microsites to provide the greatest advantage for seedlings. Although the presence or absence of microsites did not improve our models, this is likely misleading, because sites that have lower tree

densities have higher proportions of microsites. This could suggest that microsites are a primary reason why trees survive in these sites that would otherwise be unfavorable and, in better sites, the "microsite advantage" is less important.

The climate portfolio of the GYE is shifting and it will continue to shift for the foreseeable future [44]. Most notably, there is a strong warming trend with generally wetter springs, but greater water deficit by late summer [12]. Mapping these projections to the GYE landscape would simplify complexity in identifying planting sites that are suitable for establishment now and in the future.

### 4.4. Opportunities and Limitations

Limitations are largely centered on imperfect information that is associated with planting conditions. For example, we did not account for differences in the nursery stock or seed provenance of planted WBP, because we did not have this information for all sites. Therefore, each source population is likely to have some genetic variation and therefore varied response to its environment [24]. Additionally, season of planting was unknown for some sites that could affect soil moisture availability at those times that may have contributed to differential establishment among our sites. Planting trees is a skill, and each individual planter likely differed in their ability to successfully place trees in the ground, so as to promote success in a young seedling. It is also possible that planting personnel did not accurately update maps post-planting, even if they only planted on a portion of the site.

There were other limitations that were related to the observational nature of our study that we could not control. While all plantings took place post-fire, the severity and area of the fire has implications for the physical, chemical, and biological properties of the soil (seeds, fungi, bacterial communities, texture, and organic matter content) that affect growth rates, and also the rate at which competitors return to the site [24,45]. Some sites were and are still actively grazed, and the possible browsing or trampling by livestock on seedlings may disrupt WBP growth and stand density. Managers pointed out that pocket gophers (*Thomymys* genus) are often deleterious to the health of WBP, and their abundance can change from site-to-site and over time. We did not directly measure the vulnerability to drought via tissue water content that would be a more sensitive indicator of drought stress than change in height. Although growth rate was used as a response variable, it should not be interpreted as an indicator of longevity or probability of establishment. We do not know the specific cause of mortality for an individual tree, but recognize that there are agents beyond drought stress. This includes potential past episodes of blister rust.

An additional factor in our assessment of change ratio involved the inability to distinguish between naturally recruited seedlings and those that were planted. There are two ways natural regeneration could affect our study. First, it could result in a density ratio greater than one if natural regeneration added trees to planting sites at a rate that is greater than the combined mortality of planted and naturally regenerating trees. This occurred in five of 29 sites (17 percent). Second, natural regeneration could artificially increase our estimate of planted tree survival in sites that had a change ratio less than one, thus inflating our density change estimate. However, we feel the effect of natural recruitment on our interpretations was minimal in both cases for several reasons. In sites where change ratios were greater than one, we conclude these were optimum sites for establishment and growth. Climate is an important driver of tree establishment, whether naturally regenerating or planted, and there will be a range of conditions that are suitable. Outside that range neither naturally regenerated nor planted seedlings will survive. Further, if natural regeneration occurred at some sites those conditions were clearly within the range of climate suitability for establishment and growth, and it would be unlikely that those conditions were not also suitable for planted seedlings. While not conclusive, but supportive, is a study of WBP seedling establishment that found six years after direct seeding the probability of survival between six and 17 percent [46], whereas survival of WBP seedlings seven years after planting showed 53 percent survival [47]. This suggests that the probability of establishment is greater for seedlings and that direct seeding is more vulnerable to physical factors, such as high temperature that is associated with late season water deficit and biological factors, such as predation. Thus, including

natural regeneration makes our assessment of conditions conservative toward recommending lower temperature and higher rates of AET needed for establishment and growth. This is an important consideration under continued warming and drying conditions which are projected for the GYE.

Opportunities for follow-on studies include generating performance maps that are based on AET that map the interaction between climate and soil properties conducive to establishment, and landscape positions that accumulate water, such as snow drifts and convergent hillslope positions. Although we characterized the physical properties of soil that influence tree performance, future studies could also include an analysis of the nutrients that may also affect growth rate and establishment. Additionally, there are now 1244 electronically tagged individuals in historic planting sites that allow us to distinguish between existing trees and new recruits for future study. Additional confirmation of which trees were planted could be determined by shallow excavation around trunks for plastic planting containers likely still present on some of the planted individuals. Notably, it would be advantageous to combine the results from a broader range of environmental conditions that could further refine suitable biophysical environment for establishment by including additional study sites that help to better characterize natural regeneration. We suggest future studies consider the role of blister rust in seedling mortality due to the potential of unknowable blister rust mortality events of seedlings at the planting sites prior to our sampling, even though probability has been found to be greater in larger trees [48].

## 5. Conclusions

Whitebark pine seedlings must be capable of surviving environmental water stress bottlenecks after germination to become established and reach reproductive age [12]. It is prudent to prioritize localities that were buffered from intense and extended water stress, but among sites that can sustain seedlings not all are equal. Our study showed a large range in seedling growth rate correlated with actual evapotranspiration and found negative effects on the density change ratio occurred when the competitors were greater than five within a 3.59m radius of planted seedlings. Identifying the geography of advantageous climate provides information important to selecting restoration sites and understanding competition effects can inform management strategies after planting.

**Author Contributions:** The conceptualization and methodology of this project was conceived and designed by D.L., A.H., D.T., and J.H.; D.L. supervised and managed field logistics and data collection efforts; D.L. designed the electronic data sheets and completed data validation; D.L. conducted the formal analysis and wrote the paper; D.T., A.H., and J.H. helped edit and revise the paper. All authors have read and agreed to the published version of the manuscript.

**Funding:** This research was funded by USDA Forest Service's Forest Health Protection Program, University of Wyoming—National Park Service Small Grants Program, and Montana State University—Ecology Department's Rumely Award.

**Acknowledgments:** We thank our colleagues at Montana State University, Tony Chang, Kathryn Ireland, Arjun Adhikari, Kaitlin Macdonald, Kathleen Carroll, Kristen Emmett, and Kevin Barnett, were invaluable resources throughout this effort. Kristin Legg, Erin Shanahan (NPS), Robert Keane (USFS), and Jesse Logan (retired USFS) were valuable in discussions regarding GYE WBP and physiology. Additionally, Ellen Jungck, Clay Demastus, Avery Beyer, Becky Nedrow, and Kay Izlar (USFS) were helpful in providing and assisting with planting data, Ann Rodman and Zach Wilson (NPS) for expertise, and Michael Dillon, Bonnie Robinson (UW – NPS), Katie Roloson (Yellowstone Forever) and Julie Geyer (MSU) for facilitating field station and financial arrangements. Rob Daley (NPS) was instrumental in providing training to develop and utilize Esri's spatial software for use in the field. Megan Higgs and Kenneth Flagg were helpful in study design, statistical consultation, and manuscript review. Days in the field were a joy thanks to the hard work and good company of Jake Anton, Selecta Heinrich, and Florian "Flo" Heinrich.

**Conflicts of Interest:** The authors declare no conflict of interest. The funders had no role in the design of the study; in the collection, analyses, or interpretation of data; in the writing of the manuscript, or in the decision to publish the results.

## Appendix A

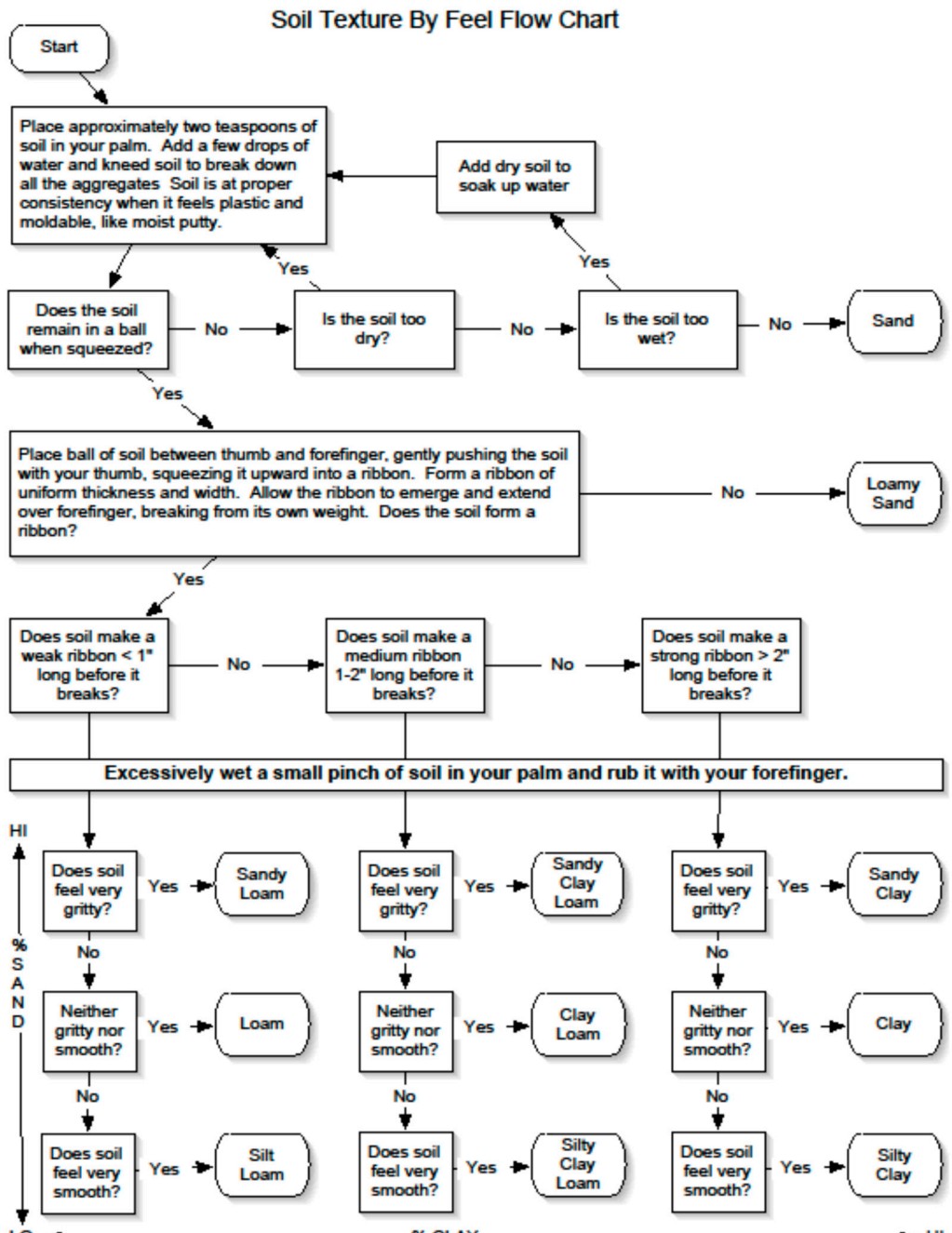

**Figure A1.** Flow chart and textural triangle used as a guide in the field to determine soil texture by feel which was used to determine field capacity of soil in the rooting zone.

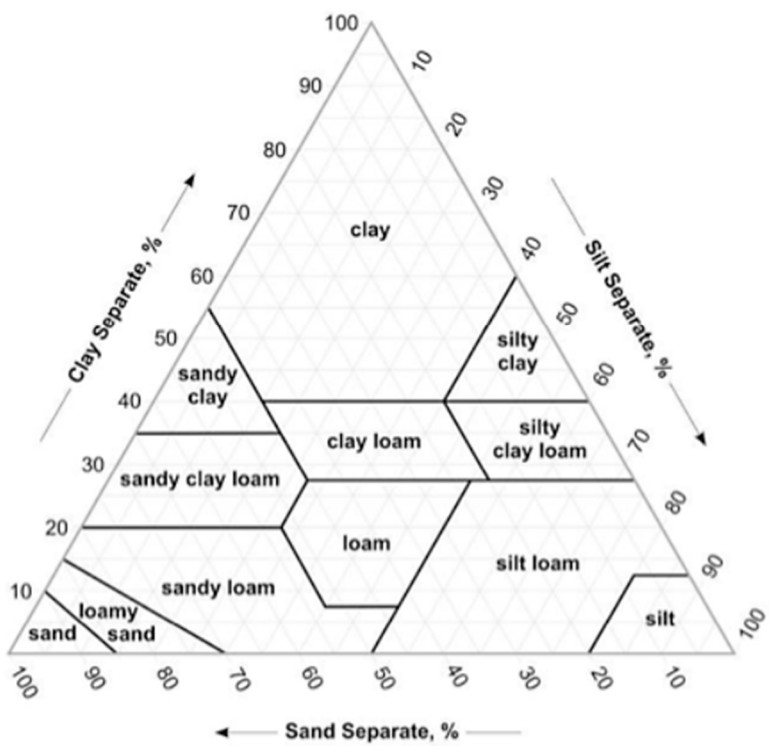

**Figure A2.** Soil textural triangle.

**Table A1.** Soil field capacity quantities in the top 203.2 mm (8 inches) of soil used for analysis in the water balance model.

| Water (mm) | Soil Type |
|---|---|
| 11.85 | Sand |
| 18.62 | Loamy sand |
| 23.70 | Sandy loam |
| 30.48 | Loam |
| 30.48 | Silt loam |
| 22.01 | Sandy Clay Loam |
| 27.09 | Sandy Clay |
| 27.09 | Clay loam |
| 32.17 | Silty clay loam |
| 40.64 | Silty Clay |

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
