# Peer review of "Biophysical Gradients and Performance of Whitebark Pine Plantings in the Greater Yellowstone Ecosystem"

_forests, doi:10.3390/f11010119_

Round 1
Reviewer 1 Report
n/a
Author Response
Done
Reviewer 2 Report
General comments
I note that the authors have responded to most of the previous reviewer comments and suggestions.
Having not seen the manuscript earlier, I have some separate issues which hopefully the authors can address.
While the topic is of interest, the methods being used have serious limitations. These may have not be created by the authors, but do create limits on what actually can be learned.
The main concern I have is that the study design does not allow one to study fundamental processes such as survivorship. The assumption seems to have been that there were no WBP seedlings present at the time of planting. That might be true and there might be reasons to suspect it was true, but the authors do not explain either the assumption or its basis. Therefore it is impossible to determine what the variables analyzed by the authors actually mean. For example, for a seedling density ratio of more than 1, does that mean that more WBP seedlings were recruited than planted? Or does that mean that there were many more seedlings on the site at the time of planting than were planted? The latter does not tell one much of meaning. The authors may have a good basis for their assumption, but it has to be specifically stated.
I was left wondering what the physiological basis for the response of height growth rate to AET would be. I found it somewhat odd that it was the two regions with low sample numbers (at least that is what the figure seems to indicate) that had the largest response. That was true for competitors. It makes me wonder if this is just a sample size effect. Certainly the uncertainty bounds leave open the possibility that there is no significant difference from the zones with higher sampling density.
It would be helpful for the authors to provide some sense of the measurement precision. For example, while I am sure that the 0.01 cm height growth rate could be computed, I have doubts the measurements could be that precise. In the field it is probably difficult to measure height to the nearest 1 cm. This also implies a rate of leader growth that seems extremely low. It might have occurred, but it would mean branches were right on top of each other. Did the authors actually see this? It would be helpful to know.
Specific comments (line)
41 Suggest that pests be replaced by insects, if that is the taxon that is causing the damage/mortality. Insects is a less pejorative word and more accurate.
95 This reads oddly given that WBP is a conifer. Maybe adding in “other” would make this clearer?
96 While I think it is a nice idea to use the idea of a fundamental niche, the authors do not describe this in the preceeding introduction. This makes it come from out of the blue (so to speak). Would it be possible to describe either the concept as it pertains to WBP or the fundamental niche of WPA?
103 A word seems to be missing here. Also here and elsewhere the words “in order” are never needed in a sentence. I recommend these two words be removed as it should not change the meaning of the sentences.
122 Not sure I completely understand the random, systematic matrix concept. Was it that systematic grids were placed in random locations?
142 I am assuming this was the length of the seedlings? The author could make this clearer.
143 Was this the length of the root wads? Or the diameter. The meaning is not clear.
157 Isn’t this the survival rate? Why use a term that does not have a transparent meaning? I am not sure that the term the authors use makes sense. It is a ratio of densities, but I don’t see how it is a change ratio. That is it not the ratio of density changes, it is the ratio of densities. That might imply some change if the ratio is less than 1. Still this is what is called survivorship.
236 Wouldn’t this be in the past tense?
252 It would help if the authors indicated this was height growth. Also it is hard to see how a growth rate of 0.01 cm could be actually measured given the methods. This is probably essentially zero growth.
280 Now I see why the authors used this terminology for the ratio of seedings at the two different times. In some cases this ratio was above zero. I may have missed this, but the WBP seedlings were only planted at sites where there had been no existing WBP seedlings? Otherwise this ratio, particularly when it exceeds 1 could be misleading.
343 What is a poor soil? Does this mean low waterholding capacity? Can the authors be more specific?
Round 2
Reviewer 2 Report
The authors have adequately addressed my concerns.
This manuscript is a resubmission of an earlier submission. The following is a list of the peer review reports and author responses from that submission.
Round 1
Reviewer 1 Report
Authors pose a very relevant question: "Our driving question is to what extent do local biophysical gradients affect GYE WBP 107 performance as represented at established planting locations?" They have a large number of planted sites (29) with the oldest at 28 years post-planting. This is an excellent source to learn from. Glacier National Park has a long history of planting. It would be useful for authors to acknowledge this and mention why their results are not considered here. The authors focus on drought-related factors. While these are likely relevant, there are other major factors that affect performance of WBP plantings, especially planting stock, disease and rodents. Blister rust is the leading cause of mortality to whitebark pine in its range, yet authors don’t describe the impact on their seedlings, thus the results are wanting. Seedlings/saplings are more vulnerable to disease-induced mortality than mature trees. Why not test as a Predictor Variable (Table 3)? There are models that report rust hazard based on biophysical gradients (possibly available for GYE). Report your survey findings in the Results (or at least address in the Discussion). Rodents are a major killer of young whitebark pine. I see no mention in methods, or even an acknowledgement. It’s surprising that only five sites had natural regeneration. Nevertheless, natural regen is not adequately addressed. The authors do not separate natural regen from planted, thus confounding results. Authors need to explain 1) proportion of regen this is natural vs. planted, 2) if natural regen was included in growth results and if so, 3) compare natural vs. planted stock. Authors do not address how different planting stock used at different sites may have translated into results. If different seedlots (parents) were used at all sites, then we would expect growth/survivorship differences. Authors need to disclose and discuss how stocks were distributed. Authors do not mention if disease-resistant stock was used, and if so, at what sites. This would make a difference in performance / survivorship. The paper would greatly benefit by adding more results. Specifically, survivorship, growth, and mortality agents could be presented. Although the dataset may not benefit from historical periodic re-measurements, the 2018 results could be expanded with more detail of the above factors. The growing number of practitioners of whitebark pine planting would eagerly welcome such insight from the GYE. Lines/Comments: 52: Perkins found much older (>1,000 years) whitebark pine in central Idaho. Figure 1: Change color of five sites to a color more visible than green. 157-159: This is very unclear. Describe what ‘digitally mapped’ actually means (e.g. what details and spatial resolution used?). 160: Unclear how plots relate to the 10m x 10m grid. Also, how many plots? Better define “near” each sapling. Was this “near” distance a constant for all plots? 168: Water holding capacity is defined in paper, but unclear how this was measured/classified. Does Table 2 determine? If so, how is it quantified? The reader would benefit by clarifying. 188: I’ve never seen the term ‘weeping’ associated with whitebark pine. This should be clarified or removed. 188-190: This does not inspire confidence that authors understand blister rust. Fruiting bodies are not the same as cankers. Squirrels need no acknowledgement …..as authors state…..they only affect older trees. 211: water storage or ‘holding’? Figure 5: Larger fonts needed for axis titles. 382: add citation 394: “species range edge” in GYE? It occurs outside GYE, so unclear.Author Response
Please see the attachment.

Reviewer 2 Report
An assessment of survival, growth and seedling health post planting is essential for refining and improving whitebark pine plant prescriptions. Announcements of this research have garnered a lot of interest in the restoration community. While the sampling design and statistics employed are commendable, there are other omitted factors impacting seedling survival that would have informed final model selection, such as soil parent material, Hargreaves climatic moisture deficit and known levels of blister rust resistance among seed sources. Blister rust infection data were collected, but no metrics were identified. These data were also not involved in any modeling; collectively, these four predictors were a missed opportunity to describe planting successes or failures. I have concerns over basic knowledge of what growth stage is impacted by mountain pine beetle and because this paper is about the seedling growth stage, why is a large portion of the manuscript devoted to mountain pine beetle impacts? It was not a variable utilized in any of the modeling. Stock type has also varied in GYE plantings, did stock type (2-yr vs. 3-yr container) have any impact on survival? These data could be evaluated as a fixed effect. If AET retains its position of significance in the final version of your manuscript, I am hopeful technology transfer in the form of spatial data will be made available to practitioners.

Round 2
Reviewer 1 Report
Overall comment:
In the revised version of this manuscript, the authors now better acknowledge the uncertainties that are inherent to this study. These include planting stocks, blister rust, natural vs. artificial regeneration. Further, they added additional variables (e.g. livestock and season of planting) as factors that can affect survivorship. While these are acknowledged, they are largely unmitigated in this new version. As such, fundamental shortcomings remain, and I don’t think the authors can isolate their findings (e.g. water holding capacity) in concluding where restoration planting will be most successful.
Additional comments:
It appears that 1,244 seedlings were evaluated. If correct, this would very likely be an inadequate sample number to characterize differences between 29 sites. This equates to only an average of 43 saplings evaluated per site and 1.2 seedlings per acre.
The numbers and densities of seedlings at original plantings is very important to determining survivorship results. Yet authors don’t convey that the records were detailed. “Estimated number of seedlings planted was divided by the estimated planting site area in order to calculate the initial WBP density at each site”. Because both seedling numbers and site area are estimates, there is a large amount of uncertainty in the final estimate used for results calculations.
Blister rust (a leading killer of whitebark pine) remains inadequately addressed. Authors state that only 3 trees were infected. This equates to 0.24% incidence of rust (3/1,244). This is very unlikely, given that the amount of trees that are infected in the GYE is 20-30%. Authors response: “Seedlings….less likely to be infected. (Thoma et al.; in prep; this issue of Forests)”. I find this very unlikely. Instead, SURVEYORS are less likely to detect rust on seedlings vs. larger trees. It’s simply more difficult to detect. Indeed, results presented may reflect blister rust incidence. The Wind River range has comparatively lower rust. Thus, the higher regen density at Wind River (Fig 6) may be a result of this more than the soil results presented.
Recommend:
Authors should use only sample sites that have detailed planting records (numbers and spatial delineations) in order to avoid ‘estimates’ of original stocking number. This will also address natural regeneration. Authors should sample more trees per site. Authors apply blister rust hazard models to each site. As of now, the paper does not account for rust mortality. Results will likely support that rust hazard is in an important factor affecting results. If results determine otherwise, this will still be valuable for addressing a critical information gap that remains in this manuscript. Alternatively, authors may re-sort analysis and results. By comparing sites within study units rather than across, this approach would better limit regional differences in blister rust and possibly other factors (e.g. climate).Reviewer 2 Report
Incomplete response to reviewer comments. The following items still need to be addressed:
Lines 121-122 Figure 1. Need to label plot locations BT = Beartooth, EC = East Centennial, WC = West Centennial, WI = Wind River, WY = West Yellowstone, so the reader can better understand/visualize your results.
Lines 140-141. A descriptive table of seed lots used with their attributes (latitude, longitude, elevation, and rust resistance level) needs to be included in the main body of the paper, or as supplemental material. Even if seed lot data are not available, supplemental material summarizing planting sites, only 29 entries, would better support data transparency.
Lines 347-353 Could soil parent material be contributing to the threshold in AET? Hargreave’s climatic moisture deficit may have been an informative predictor in conjunction with AET. Was this climate variable considered, see Lutz et al. 2010.
Lines 391-392 in first version and lines 426-427 in revised manuscript. The following statement is false “This is a major consideration for any planting efforts that plant seedlings that require a century of growth prior to production of seeds.” It is well-documented that WBP in the GYE begins cone production between ages 50-80. Range-wide, there is nowhere in the species distribution where WBP must attain an age of 100 years before seed production begins. Change century to 50-80 years.
Line 117 change 40 decades to 30 decades. The first planting of WBP in the GYE occurred on the Custer-Gallatin NF in 1988. And line 125 in the revised manuscript.
Revised manuscript.
Lines 381-383 awkward language suggest ( ) or eliminating planting locations altogether, ..."Notably, out of 1,244 trees sampled, only three individual WBP showed signs of white pine blister rust (two trees in the Beartooths and one tree in the West Centennials Planting Units).
Lines 455-488 needs more attention to grammar, sentence structure and excessive word count (reads as if hastily put together).
Line 456. Consider addition of a sentence introducing other factors/considerations not addressed in your study that could have contributed to your results and interpretation of those data.
Line 458 suggest replacing "some variation of genetics" to "some genetic variation"
Lines 459-460 suggest replacing current sentence with ... season of planting (spring vs. fall), available precipitation and soil moisture at those times may have contributed to differential survival among our sites.
Lines 465 omit reference to Pinus contorta altogether or add other competing species such as Engelmann spruce and subalpine fire. What is biological significance of serotinous cones? Lodgepole has both serotinous and open-faced cones in the GYE. Can the authors provide a citation that natural regeneration from serotinuous cones exceeds that of open-faced cones, thereby contributing to competition?
Line 460-462, 465-466 omit supposition regarding planting skills and data management.
Line 475 omit supposition regarding blister rust. With an occurrence of only 3 trees (0.24%), even the authors data support this disturbance agent was not statistically significant for these planting sites.
Line 487 omit reference to Glacier National Park altogether or include the other ecosystems where planting has occurred, i.e., Central Montana And Bitterro